

# Weak-constraint inverse modeling using HYSPLIT Lagrangian dispersion model and Cross Appalachian Tracer Experiment (CAPTEX) observations – Effect of including model uncertainties on source term estimation

Tianfeng Chai[1,2], Ariel Stein[1], and Fong Ngan[1,2]

[1]NOAA Air Resources Laboratory (ARL), NOAA Center for Weather and Climate Prediction, 5830 University Research Court, College Park, MD 20740, USA

[2]Cooperative Institute for Climate and Satellites, University of Maryland, College Park, MD 20740, USA

**Correspondence:** Tianfeng Chai (Tianfeng.Chai@noaa.gov)

**Abstract.** A HYSPLIT inverse system that is based on variational data assimilation and a Lagrangian dispersion transfer coefficient matrix (TCM) is evaluated using the Cross Appalachian Tracer Experiment (CAPTEX) data collected from six controlled releases. For simplicity, the initial tests are applied to release 2 for which the HYSPLIT has the best performance. Before introducing model uncertainty terms, the tests using concentration differences in the cost function results in severe underestimation

while those using logarithm concentrations differences results in overestimation of the release rate. Adding model uncertainty terms improves results for both choices of the metric variables in the cost function. A cost function normalization scheme is later introduced to avoid spurious minimal source term solutions when using logarithm concentration differences. The scheme is effective in eliminating the spurious solutions and it also helps to improve the release estimates for both choices of the metric variables. The tests also show that calculating logarithm concentration differences generally yield better results than calculating

concentration differences and the estimates are more robust for a reasonable range of model uncertainty parameters. This is further confirmed with nine ensemble HYSPLIT runs in which meteorological fields were generated with varying planetary boundary layer (PBL) schemes. In addition, it is found that the emission estimate using a combined TCM by taking the average or median values of the nine TCMs is similar to the median of the nine estimates using each of the TCMs individually. The inverse system is then applied to the other CAPTEX releases with a fixed set of observational and model uncertainty param-

eters and the largest relative error among the six releases is $53.3\%$. At last, the system is tested for its capability to find a single source location as well as its source strength. In these tests, the location and strength that yield the best match between the predicted and the observed concentrations are considered as the inverse modeling results. The estimated release rates are mostly not as good as the cases in which the exact release locations are assumed known, but they are all within a factor of 3 for all the six releases. However, the estimated location may have large errors.



# 1 Introduction

The transport and dispersion of gaseous and particulate pollutants are often simulated to generate pollution forecasts for emergency responses or produce comprehensive analyses of the past for better understanding of the particular events. Lagrangian particle dispersion models are particularly suited to provide plume products associated with emergency response scenarios.

While accurate air pollutant source terms are crucial for the quantitative predictions, they are rarely provided in most applications and have to be approximated with a lot of assumptions. For instance, the smoke forecasts over the continental U.S. operated by the National Oceanic and Atmospheric Administration (NOAA) using the HYSPLIT model (Draxler and Hess, 1997; Stein et al., 2015) in support of the National Air Quality Forecast Capability (NAQFC) relies on the outdated fuel loadings data and a series of assumptions related to smoke release heights and strength approximation (Rolph et al., 2009).

Observed concentration, deposition, or other functions of the atmospheric pollutants such as aerosol optical thickness measured by satellite instruments can be used to estimate some combination of source location, strength, and temporal evolution using various source term estimation (STE) methods (Bieringer et al., 2017; Hutchinson et al., 2017). Among the applications, the recent Fukushima Dai-ichi Nuclear Power Plant accidents saw the most implementations of the STE methods to estimate the radionuclide releases. The STE methods range from simple comparisons between model outputs and measurements

(e.g. Chino et al., 2011; Katata et al., 2012; Terada et al., 2012; Hirao et al., 2013; Kobayashi et al., 2013; Oza et al., 2013; Katata et al., 2015; Achim et al., 2014) to those sophisticated ones using various dispersion models and inverse modeling schemes (e.g. Stohl et al., 2012; Winiarek et al., 2012; Saunier et al., 2013; Winiarek et al., 2014; Chai et al., 2015). Another active field for STE applications is the estimation of the volcanic ash emissions. Many attempts have been made for several major volcano eruptions (Wen and Rose, 1994; Prata and Grant, 2001; Wilkins et al., 2014, 2016; Chai et al., 2017).

While there are many STE methods applied to reconstruct the emission terms, it is still a state of art. Two popular advanced inverse modeling approaches are cost-function-based optimization methods and those based on Bayesian inference. However, it is difficult to evaluate the STE without knowing the actual sources for most applications. Chai et al. (2015) generated pseudo observations using the same dispersion model in their initial inverse experiment tests, which are often called "twin experiments". Such tests allow observational errors to be added realistically (e.g. Chai et al., 2015), but it is non-trivial to represent

the model errors incurred by other model parameters such as the uncertainties of the meteorological field. One way to objectively evaluate the inverse modeling results is to compare the predictions with the independent observations or withheld data. However, such indirect comparisons still cannot provide quantitative error statistics for the source terms.

There have been some tracer experiments conducted to study the atmospheric transport and dispersion with controlled releases. In these experiments, the source terms were well-quantified and comprehensive measurements were made subsequently

over an extended area (e.g. Draxler et al., 1991; Van Dop et al., 1998). With the known source terms, they provide a unique opportunity to evaluate the STE methods. Singh and Rani (2014) and Singh et al. (2015) used measurements from a recent dispersion experiment (Fusion Field Trials 2007) data to evaluate a least-squares technique for identification of a point release. The European Tracer Experiment (ETEX) data set were also used to study the STE methods based on the principle of maximum





entropy and a least squares cost function (Bocquet, 2005, 2007, 2008). However, such formal evaluation of the STE methods is still very limited.

A HYSPLIT inverse system based on 4D-Var data assimilation and a transfer coefficient matrix (TCM) was developed and applied to estimate cesium-137 source from the Fukushima nuclear accident using air concentration measurements (Chai et al., 2015). The system was further developed to estimate the effective volcanic ash release rates as a function of time and height by assimilating satellite mass loadings and ash cloud top heights (Chai et al., 2017). In this study, the Cross Appalachian Tracer Experiment (CAPTEX) data are used to evaluate the HYSPLIT inverse modeling system. The paper is organized as follows. Section 2 describes the CAPTEX experiment, HYSPLIT model configuration, and the source term inversion method. Section 3 presents emission inversion results and a summary is given in Section 4.

## 2   Method

### 2.1   CAPTEX experiment

The CAPTEX experiment consisted of seven near-surface releases of the inert tracer perfluro-monomethylcyclohexane (PMCH) from Dayton, Ohio, U.S. and Sudbury, Ontario, Canada during September and October 1983 (Draxler, 1987). Table 1 lists the locations, time, amounts, and measurement counts of the seven releases. Samples were collected at 84 different measurement sites distributed from 300 to 1100 km downwind of the emission source, as either 3- or 6-hour averages up to 60 hours after each release. Figure 1 shows the distribution of measurement sites and the two source locations. Since there were few measurements above twice background values for release 6, it will be excluded from the testing as in the earlier studies using CAPTEX data (e.g. Hegarty et al., 2013; Ngan et al., 2015). Note that 3.4 fl/l has been subtracted from all CAPTEX measurements to remove background and "noise" in sampling where the ambient background concentration is constant at 3.0 fl/l (Ferber et al., 1986). At ground level, 1 fl/l is equivalent to 15.6 $pg/m^3$. Duplicate sample analyses showed that the majority data has a mean standard deviation estimated as $10.8\%$ but contaminated samples may have standard deviation as large as $65\%$ (Ferber et al., 1986).

### 2.2   HYSPLIT

In this study, the tracer transport and dispersion are modeled using the HYSPLIT model in its particle mode in which three-dimensional (3D) Lagrangian particles released from the source location passively follow the wind field (Draxler and Hess, 1997, 1998; Stein et al., 2015). A particle release rate of 50,000 per hour is used for all calculations. Random velocity components based on local stability conditions are added to the mean advection velocity in the three wind component directions. The meteorological data used to drive the HYSPLIT are time-averaged from the Advanced Research WRF model (ARW, version 3.2.1) simulation results at 10-km resolution and they are identical to those used by Hegarty et al. (2013). The 10-km run was nested inside a larger domain at 30-km resolution, over which the simulation was started using the North American Regional Reanalysis (NARR) at 32-km (Mesinger et al., 2006). In the WRF simulations, 3D grid nudging of winds was applied in the



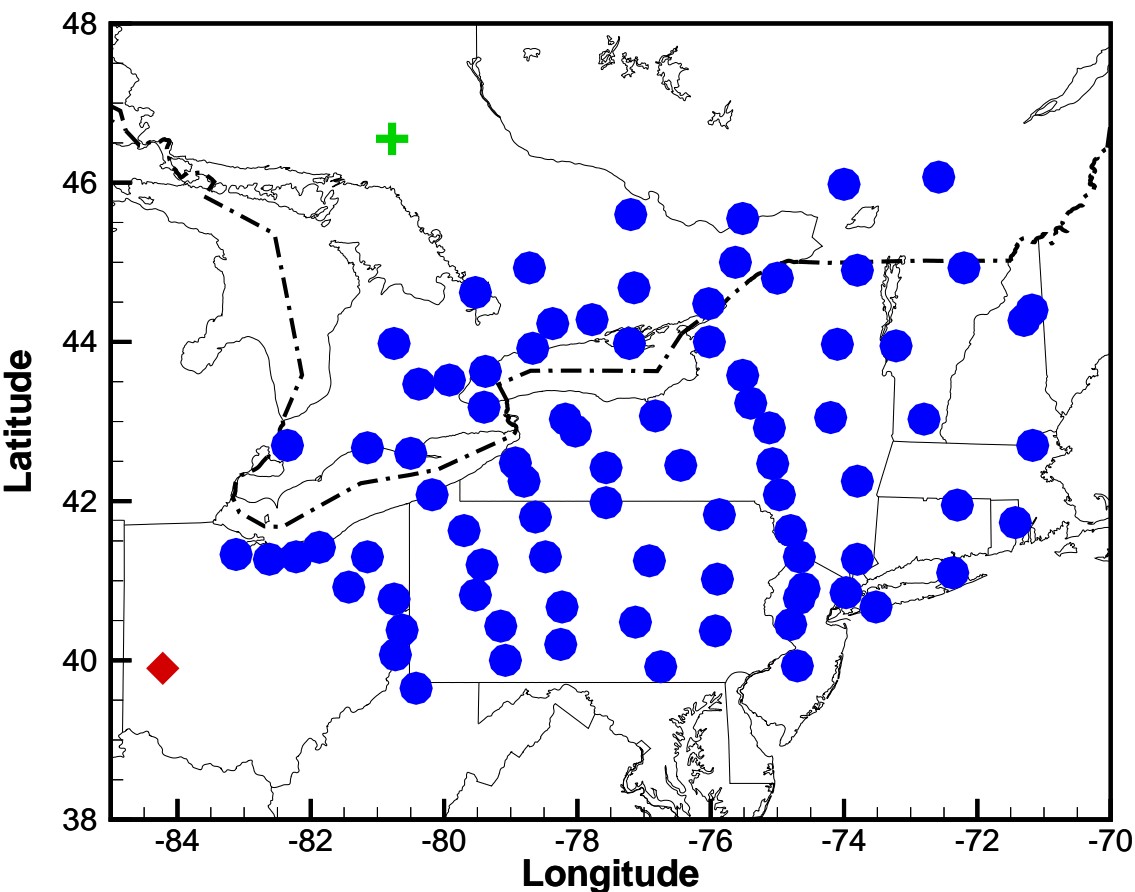

**Figure 1.** Distribution of the 84 measurement sites and two CAPTEX source locations (Dayton, Ohio, U.S. shown as a red diamond, and Sudbury, Ontario, Canada shown as a green cross).





**Table 1.** The locations, time, amounts, and measurement counts ($M_{obs}$) of each CAPTEX release from Dayton, Ohio, U.S. and Sudbury, Ontario, Canada during September and October 1983.

| # | Site (latitude, longitude) | Release time | Amount | $M_{obs}$ |
|---|---|---|---|---|
| 1 | Dayton (39.80°, -84.05°) | 1700-2000Z, Sep. 18, 1983 | 208 kg | 395 |
| 2 | Dayton (39.90°, -84.22°) | 1705-2005Z, Sep. 25, 1983 | 201 kg | 400 |
| 3 | Dayton (39.90°, -84.22°) | 1900-2200Z, Oct. 02, 1983 | 201 kg | 404 |
| 4 | Dayton (39.90°, -84.22°) | 1600-1900Z, Oct. 14, 1983 | 199 kg | 367 |
| 5 | Sudbury (46.62°, -80.78°) | 0345-0645Z, Oct. 26, 1983 | 180 kg | 357 |
| 6 | Dayton (39.90°, -84.22°) | 1530-1600Z, Oct. 28, 1983 | 32 kg | - |
| 7 | Sudbury (46.62°, -80.78°) | 0600-0900Z, Oct. 29, 1983 | 183 kg | 358 |

free troposphere and within the planetary boundary layer (PBL). There are 43 vertical layers with the lowest one being approximately 33 m thick. Tracer concentrations are computed over each grid cell by summing the mass of all particles in the cell and dividing the result by the cell's volume. In this study, the concentration grid cells have $0.25^o$ resolution in both latitude and longitude directions and vertically they extend 100 m from the ground.

To avoid running the HYSPLIT modeling repeatedly, a TCM is generated similar to the previous HYSPLIT inverse modeling studies (Chai et al., 2015, 2017). As described in Draxler and Rolph (2012), independent simulations are performed with a unit emission rate from each source location and a pre-defined time segment. Each release scenario is simply a linear combination of the unit emission runs.

### 2.3 Emission Inversion

Similar to Chai et al. (2015), the unknown releases can be solved by minimizing a cost functional that integrates the differences between model predictions and observations, deviations of the final solution from the first guess (*a priori*), as well as other relevant information written into penalty terms (Daley, 1991). For the current application, the cost functional $\mathcal{F}$ is defined as,

$$\mathcal{F} = \frac{1}{2}\sum_{i=1}^{M}\sum_{j=1}^{N}\frac{(q_{ij}-q_{ij}^b)^2}{\sigma_{ij}^2} + \frac{1}{2}\sum_{m=1}^{M}\frac{(c_m^h-c_m^o)^2}{\epsilon_m^2} + \frac{c_{sm}}{2}\cdot\sum_{i=2}^{N-1}[\frac{(q_{i-1,j}-q_{i-1,j}^b)-2\cdot(q_{ij}-q_{ij}^b)+(q_{i+1,j}-q_{i+1,j}^b)}{q_c}]^2 \quad (1)$$

where $q_{ij}$ is the discretized source term at hour $i$ and location $j$ for which an independent HYSPLIT simulation has been
run and recorded in a TCM. $q_{ij}^b$ is the first guess or *a priori* estimate and $\sigma_{ij}^2$ is the corresponding error variance. Note that all tracer sources in this study were at ground level and the release heights in the HYSPLIT were set as 10 m for all the following test cases. We also assume the uncertainties of the release at each time-location are independent of each other so that only the diagonal term of the typical *a priori* error variance $\sigma_{ij}^2$ appears in Equation 1. $c^h$ and $c^o$ denote HYSPLIT-predicted and measured concentrations, respectively. The observational errors $\epsilon_m^2$ are assumed to be uncorrelated. As the term $\epsilon_m^2$ is
essentially used to weight $(c_m^h-c_m^o)^2$ terms, the uncertainties of the model predictions and the representative errors should be



included besides the observational uncertainties. This will be further discussed in Section 3.2. The last term is a smoothness penalty to make the modified minimization problem better conditioned (Lin et al., 2002). $q_c$ is a scale constant and may be combined with $c_{sm}$ to adjust the smoothness term. In this study, the smoothness penalty is turned off by setting $c_{sm}$ as zero. A large-scale bound-constrained limited-memory quasi-Newton code, L-BFGS-B (Zhu et al., 1997) is used to minimize the

cost functional $\mathcal{F}$ defined in Equation 1 when multiple parameters need to be determined. As shown by Chai et al. (2015), the metric variable can be changed from concentration to logarithm concentration. Both choices of metric variable will be tested here.

## 3   Results

### 3.1   Recovering emission strength without model uncertainty

As an initial test, the exact release location and time are both assumed known and the only unknown variable left to be determined is the release rate, or the total release amount. For this type of one-dimensional problem, an optimal emission strength can be easily found without having to use sophisticated minimization routines. For instance, the $\mathcal{F}$ may be directly calculated for a number of emission strength values and the resulting $\mathcal{F} = \mathcal{F}(q)$ plot will reveal the optimal $q$ strength that is associated with the minimal $\mathcal{F}$. Note that such an optimal solution not only depends on the chosen parameters in Equation 1,

but also highly depends on the HYSPLIT model setup and the meteorological fields.

Both Hegarty et al. (2013) and Ngan et al. (2015) showed that the HYSPLIT dispersion model performed better for Release 2 than the other releases. Thus Release 2 is initially chosen to perform a series of inverse modeling tests. Assuming no prior knowledge of the emission strength, the first guess is given as $q^b = 0$, and $\sigma = 10^4 \ kg/hr$ is assumed. Sensitivity tests show that when $q^b$ is changed to 100 $kg/hr$ the emission strength estimates are nearly unchanged with the same or larger $\sigma$.

Firstly, the observational uncertainties are formulated to include a fractional component $f^o \times c^o$ and an additive part $a^o$. No model uncertainties are considered to contribute to $\epsilon$. Table 2 lists the emission strength $q$ that generates the minimal cost function for a series of $f^o$ and $a^o$ combinations, where $f^o$ ranges from 10% to 50%, and $a^o$ is assigned as 10, 20, and 50 $pg/m^3$. All the emission strength values obtained are significantly lower than the actual release of 67 $\ kg/hr$. It shows that a larger $f^o$ value tends to have a smaller $q$ estimate, but a larger $a^o$ results in a larger $q$. The significant underestimation of

the release strength is caused by the implicit assumption of a perfect model when $\epsilon$ does not include the model uncertainties. Figure 2 shows the comparison between the predicted and measured concentrations when the actual release rate of 67 $\ kg/hr$ is applied. Large discrepancies still exist even when the exact release is known and used in the simulation. For the measured zero concentrations, most of the predicted values are non-zero and can be above 1000 $pg/m^3$. As $\epsilon_m = a^o$ for these zero concentrations, $\frac{(c_m^h - c_m^o)^2}{\epsilon_m^2}$ will dominate the cost function when $a^o$ is not large enough. This explains that the underestimation

is not as severe for $a^o$ =50 $pg/m^3$ as that for $a^o$ =10 $pg/m^3$. While $\epsilon$ do not change with $f^o$ for the zero concentrations, smaller $f^o$ values help increase the weighting of the terms $\frac{(c_m^h - c_m^o)^2}{\epsilon_m^2}$ associated with large measured concentrations. So, the estimated emission strength when $f^o = 10\%$ is better than when $f^o = 50\%$.




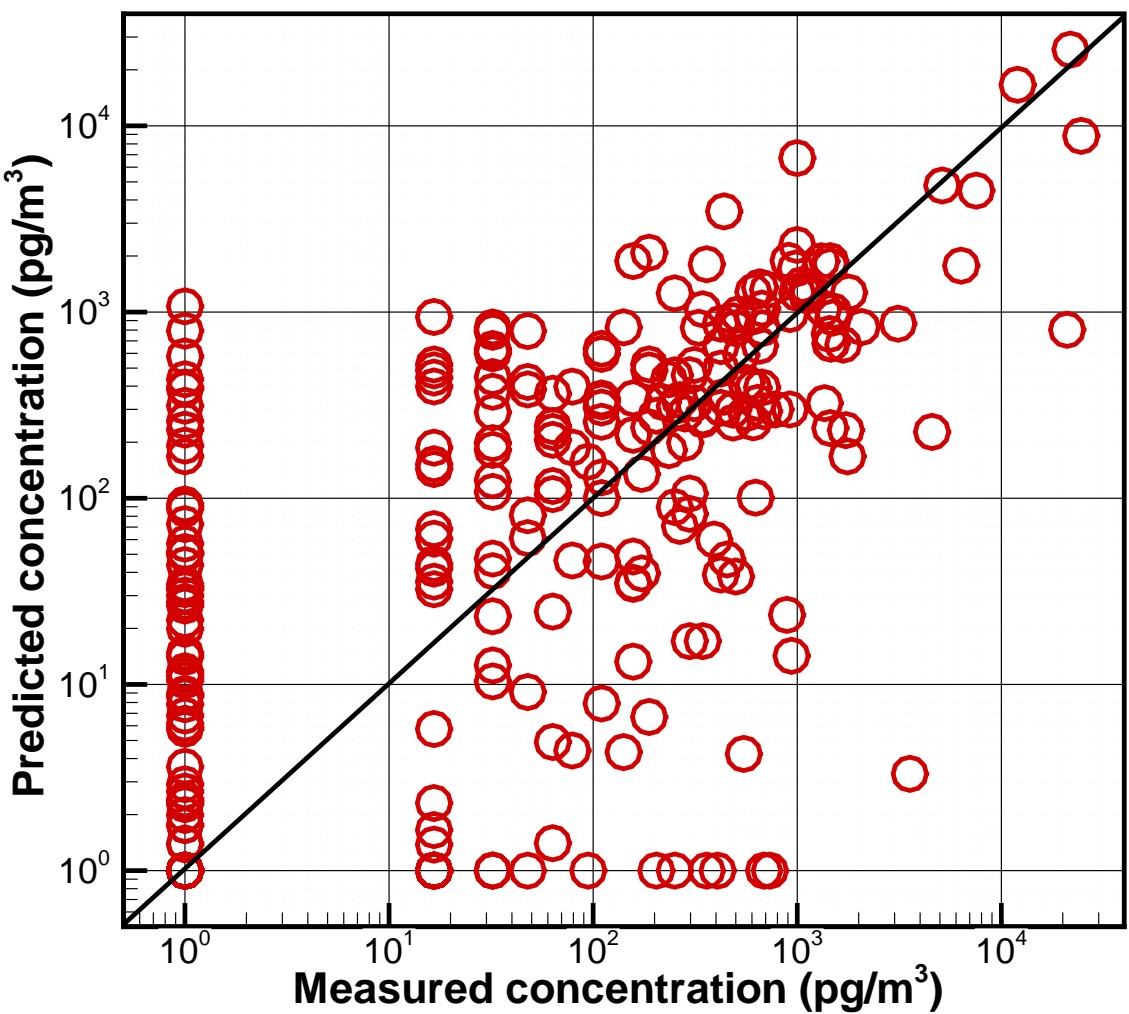

**Figure 2.** Comparison between the predicted and measured concentrations for Release 2 during the CAPTEX experiment. In the HYSPLIT simulation, at the exact release location, an emission rate of 67 $kg/hr$ was applied from 17Z to 20Z on September 25, 1983. A constant 1 $pg/m^3$ is added to both predicted and measured concentrations to allow logarithm calculation.





As stated in Chai et al. (2015), the metric variable in Equation 1 can be changed to $ln(c)$, i.e. replacing $(c_m^h - c_m^o)$ with $ln(c_m^h) - ln(c_m^o)$. A constant $0.001\ pg/m^3$ is added to both $c_m^h$ and $c_m^o$ to allow the logarithm operation for zero concentrations. In such a case, $\epsilon_m^{ln(c)}$ can be calculated as

$$\epsilon_m^{ln(c)} = ln(1 + f^o + \frac{a^o}{c_m^o}) \tag{2}$$

5    Note that $0.001\ pg/m^3$ is also added to $c_m^o$ in the second term to avoid dividing by zero. The $\frac{a^o}{c_m^o}$ term in Equation 2 makes $\epsilon_m^{ln(c)}$ larger for measured low concentrations than those measured high concentrations, thus makes the measured zero concentrations have little effect in the final emission strength estimates. Table 3 shows that the emission strengths are overestimated, but are within a factor of 2 over the actual release of $67\ kg/hr$, for all $f^o$ and $a^o$ combinations. The similar trends of how $q$ changes with $f^o$ and $a^o$ are also observed here, i.e., a larger $a^o$ or a smaller $f^o$ tends to have a larger $q$ estimate.

**Table 2.** Emission strength of release 2 that minimizes $\mathcal{F}$ for different observational errors, defined as $\epsilon = f^o \times c^o + a^o$. Concentration is used as the metric variable.

| Emission $(kg/hr)$ | $a^o$ =10 $pg/m^3$ | $a^o$ =20 $pg/m^3$ | $a^o$ =50 $pg/m^3$ |
|---|---|---|---|
| $f^o = 10\%$ | 7.1 | 11.1 | 17.4 |
| $f^o = 20\%$ | 4.1 | 7.1 | 12.6 |
| $f^o = 30\%$ | 2.9 | 5.2 | 10.0 |
| $f^o = 50\%$ | 1.8 | 3.4 | 7.1 |

**Table 3.** Emission strength of release 2 that minimizes $\mathcal{F}$ for different observational errors, defined as $\epsilon = f^o \times c + a^o$. Logarithm concentration is chosen as the metric variable, i.e. $(c_m^h - c_m^o)$ in Equation 1 is replaced with $ln(c_m^h) - ln(c_m^o)$.

| Emission $(kg/hr)$ | $a^o$ =10 $pg/m^3$ | $a^o$ =20 $pg/m^3$ | $a^o$ =50 $pg/m^3$ |
|---|---|---|---|
| $f^o = 10\%$ | 115.2 | 119.8 | 124.7 |
| $f^o = 20\%$ | 106.3 | 112.9 | 119.8 |
| $f^o = 30\%$ | 101.2 | 108.5 | 116.3 |
| $f^o = 50\%$ | 94.4 | 101.2 | 109.6 |

10    While using logarithm concentration as the metric variable yields better emission estimates than using concentration as the metric variable, the results in Table 3 are apparently systematically overestimated, comparing to the systematically underestimated results in Table 2. In addition, the $f^o$ and $a^o$ combinations associated with the best emission estimates in Tables 2 and 3 appear to be in the opposite corners of the tables.





## 3.2 Recovering emission strength with model uncertainty

To consider the model uncertainties in a simplified way, $\epsilon^2$ will be formulated as

$$\epsilon_m^2 = (f^o \times c_m^o + a^o)^2 + (f^h \times c_m^h + a^h)^2 \tag{3}$$

As $a^o$ and $a^h$ affect the $\epsilon^2$ in a similar way, the representative errors caused by comparing the measurements with the predicted concentrations averaged in a grid can be included in either $a^h$ or $a^o$.

With logarithm concentration as the metric variable, $(\epsilon_m^{ln(c)})^2$ is comprised of two parts, as

$$(\epsilon_m^{ln(c)})^2 = [ln(1 + f^o + \frac{a^o}{c_m^o})]^2 + [ln(1 + f^h + \frac{a^h}{c_m^h})]^2 \tag{4}$$

Note that a constant small number $0.001\ pg/m^3$ is added to denominators $c_m^o$ and $c_m^h$ to avoid dividing by zero.

Using concentration and logarithm concentration as the metric variable, respectively, Tables 4 and 5 show the emission strength estimates with different $f^h$ and $a^h$, while keeping $f^o = 20\%$, $a^o = 20\ pg/m^3$. It should be noted that the model uncertainties are not equivalent to model errors. Although dispersion model simulations can have large errors due to various reasons including the source term uncertainties, the model uncertainties are used to indicate that the model is not perfect even with the "optimal" model parameters. Similar to weak constraint applied in operational 4D-Var data assimilation systems (Zupanski, 1997; Tremolet, 2006), introducing model uncertainties is mainly intended to relax the model constraint for imperfect models. Here the $f^h$ and $a^h$ parameters are given similar ranges as those given to the observational uncertainty parameters.

When concentration is used as the metric variable, the emission strength estimates with model uncertainties considered are improved over those without model uncertainties. The estimates of emission strength generally increases with the model uncertainty, either through $a^h$ or $f^h$ except for $f^h = 50\%$, when the $q$ estimates slowly decreases with $a^h$. When $f^h = 0\%$, $a^h = 10, 20$, and $50\ pg/m^3$ while $a^o = 20\ pg/m^3$, the $q$ estimates, 7.7, 9.1, and 13.6 $kg/hr$, are inline with the results shown in Table 2, where $q = 7.1\ kg/hr$ for $a^o = 20\ pg/m^3$ and $q = 12.6\ kg/hr$ for $a^o = 50\ pg/m^3$. However, the trend of how $q$ estimates change with $f^h$ is opposite to how $q$ estimates change with $f^o$. Table 4 shows that the emission strength increases with the model uncertainty factor $f^h$. With $f^h = 20\%$, the release estimates of 48.5, 50.4, and 53.5 $kg/hr$ are all within $30\%$ of the actual release rate of 67 $kg/hr$. Instead of underestimation shown in Table 2, the release estimates are overestimated when $f^h = 50\%$ is assumed.

With logarithm concentration as the metric variable, larger $a^h$ or $f^h$ results in slightly smaller $q$ estimates. While how $q$ estimates change with $f^h$ is similar to how they change with $f^a$ in Table 3, how $q$ estimates change with $a^h$ is opposite to how $q$ estimates change with $a^o$ before introducing model uncertainties. Equation 4 shows that $f^o$ and $f^h$ affect $(\epsilon_m^{ln(c)})^2$ in a simple monotonic way, while the effect of $a_m^h$ is complicated as it is divided by the $c_m^h$ value that varies with the source terms. Table 5 shows that the source terms are no longer overestimated as those in Table 3. In fact, all cases have slight to moderate underestimation, with the worst results being $q = 42.6\ kg/hr$ when $f^h = 50\%$ and $a^h = 50\ pg/m^3$. Another aspect of using logarithm concentration as the metric variable is that the range of the release estimates are not as large as those using concentration as the metric variable.



**Table 4.** Emission strength of release 2 that minimizes $\mathcal{F}$ for different $f^h$ and $a^h$. Concentration is taken as the metric variable. $\epsilon^2 = (f^o \times c^o + a^o)^2 + (f^h \times c^h + a^h)^2$. $f^o = 20\%$, $a^o =$20 $pg/m^3$.

| Emission $(kg/hr)$ | $a^h =$10 $pg/m^3$ | $a^h =$20 $pg/m^3$ | $a^h =$50 $pg/m^3$ |
|---|---|---|---|
| $f^h = 0$ | 7.7 | 9.1 | 13.6 |
| $f^h = 10\%$ | 15.9 | 22.1 | 32.9 |
| $f^h = 20\%$ | 48.5 | 50.4 | 53.5 |
| $f^h = 50\%$ | 114.0 | 111.8 | 104.3 |

**Table 5.** Emission strength of release 2 that minimizes $\mathcal{F}$ for different $f^h$ and $a^h$. Logarithm concentration is taken as the metric variable. $(\epsilon_m^{ln(c)})^2 = [ln(1 + f^o + \frac{a^o}{c_m^o})]^2 + [ln(1 + f^h + \frac{a^h}{c_m^h})]^2$. $f^o = 20\%$, $a^o =$20 $pg/m^3$.

| Emission $(kg/hr)$ | $a^h =$10 $pg/m^3$ | $a^h =$20 $pg/m^3$ | $a^h =$50 $pg/m^3$ |
|---|---|---|---|
| $f^h = 0$ | 64.7 | 58.5 | 53.5 |
| $f^h = 10\%$ | 61.5 | 55.7 | 49.4 |
| $f^h = 20\%$ | 58.5 | 53.0 | 46.6 |
| $f^h = 50\%$ | 55.1 | 49.4 | 42.6 |

## 3.3 Cost function normalization

Without model uncertainties, the weighting terms for each model-observation pair do not change with emission estimates. When $\epsilon_m^2$ and $(\epsilon_m^{ln(c)})^2$ are formulated as in Equations 3 and 4, respectively, they vary with emission estimates. This may cause complication in some circumstances when logarithm concentration is used as the metric variable. Figure 3 shows the cost function as a function of source strength when $(\epsilon_m^{ln(c)})^2$ is defined as in Equation 4, with $f^h = 0$, $a^h =$50 $pg/m^3$, $f^o = 10\%$, $a^o =$20 $pg/m^3$. Before introducing cost function normalization, a global minimal cost function appears when release strength approaches zero while a local minimal cost function exists at 56.8 $kg/hr$. Several such instances were found when $a^h = $50 $pg/m^3$ and when $f^h$ is 0 or 10%, while both $f^o$ and $a^o$ are relatively small. The smaller cost function when release strength approaches zero is due to the increasing $(\epsilon_m^{ln(c)})^2$ in Equation 4 as $c_m^h$ gets smaller. While the model-observation differences are not smaller for lower release strength, the drastic increase of $(\epsilon_m^{ln(c)})^2$ when $a^h = 50$ $pg/m^3$ and $f^h$ is 0 or 10% results in smaller cost function with decreasing source strength. To avoid having zero source as a global minimizer in such situations, the total weighted mismatch between model simulation and observations is normalized by the total weights when $q_{ij} = q_{ij}^b$, as shown in Equation 5.





$$
\begin{aligned}
\mathcal{F} = \quad & \frac{1}{2} \sum_{i=1}^{M} \sum_{j=1}^{N} \frac{(q_{ij} - q_{ij}^b)^2}{\sigma_{ij}^2} + \frac{1}{2} \sum_{m=1}^{M} \frac{(c_m^h - c_m^o)^2}{\epsilon_m^2} \times \frac{\sum_{m=1}^{M} \frac{1}{\epsilon_m^{b\,2}}}{\sum_{m=1}^{M} \frac{1}{\epsilon_m^2}} \\
+ \quad & \frac{c_{sm}}{2} \cdot \sum_{i=2}^{N-1} \left[ \frac{(q_{i-1,j} - q_{i-1,j}^b) - 2 \cdot (q_{ij} - q_{ij}^b) + (q_{i+1,j} - q_{i+1,j}^b)}{q_c} \right]^2
\end{aligned}
\tag{5}
$$

Figure 3 shows that the cost function has the minimum at q=67.3 $kg/hr$ after normalization. Note that the dramatic difference of the cost function magnitude before and after the normalization is due to the extreme small value of $\sum_{m=1} \frac{1}{\epsilon_m^{b\,2}}$ calculated at $q_b = 0$. Tables 6 and 7 show the emission strength estimates after cost function normalization with different $f^h$ and $a^h$, while keeping $f^o = 20\%$, $a^o = 20\,pg/m^3$, using concentration and logarithm concentration as the metric variables, respectively. Note that $f^o = 20\%$ was chosen for the cases listed in Table 7, while $f^o = 10\%$ was chosen in Figure 3 to illustrate the potential problem. How estimates change with $f^h$ and $a^h$ in Tables 6 and 7 is similar to what is shown in Tables 4 and 5. The estimates are generally closer to the actual release than those obtained without the cost function normalization.

When having concentration as the metric variable and with $f^h = 50\%$, the emission strength estimates are 64.7, 64.7, and 65.3 $kg/hr$ for $a^h$=10, 20, 50 $pg/m^3$, respectively. They are all within 5% of the actual release rate. However, $f^h$ less than or equal to 20% results in significant underestimation. When having logarithm concentration as the metric variable, the source term estimates are not very sensitive to $f^h$ and $a^h$ values and the results listed in Table 7 are all withing 20% of the actual release rate. Among those estimates, a result of 67.3 $kg/hr$ when $f^h = 10\%$ and $a^h$=10 $pg/m^3$ is almost identical to the actual release rate.

**Table 6.** Emission strength of release 2 that minimizes normalized $\mathcal{F}$ defined in Equation 5 for different $f^h$ and $a^h$. Concentration is taken as the metric variable. $\epsilon^2 = (f^o \times c^o + a^o)^2 + (f^h \times c^h + a^h)^2$. $f^o = 20\%$, $a^o = 20\,pg/m^3$.

| Emission $(kg/hr)$ | $a^h$ =10 $pg/m^3$ | $a^h$ =20 $pg/m^3$ | $a^h$ =50 $pg/m^3$ |
|:---:|:---:|:---:|:---:|
| $f^h = 0$ | 7.7 | 9.1 | 13.6 |
| $f^h = 10\%$ | 10.9 | 15.1 | 26.4 |
| $f^h = 20\%$ | 32.9 | 35.6 | 41.3 |
| $f^h = 50\%$ | 64.7 | 64.7 | 65.3 |

### 3.4 Ensemble

Ngan and Stein (2017) simulated CAPTEX releases using a variety of planetary boundary layer (PBL) schemes. In their configuration, WRF version 3.5.1 was used with 27-km grid spacing and 33 vertical layers. The NARR data set was used for the initial conditions and lateral boundary conditions. The WRF model was initialized every day at 0600 UTC, and the first 18 hours of spin-up time in the 42-hour simulation were discarded. The PBL schemes used to create the WRF ensemble were the Yonsei University (Hong et al., 2006, YSU)), Mellor-Yamada-Janjic (Janjic, 1994, MYJ), Quasi-Normal Scale Elimination







**Figure 3.** Cost function as a function of source strength when $(\epsilon_m^{ln(c)})^2$ is defined as in Equation 4 before and after cost function normalization, with $f^h = 0$, $a^h =$50 $pg/m^3$, $f^o = 10\%$, and $a^o =$20 $pg/m^3$.





**Table 7.** Emission strength of release 2 that minimizes normalized $\mathcal{F}$ defined in Equation 5 for different $f^h$ and $a^h$. Logarithm concentration is taken as the metric variable. $(\epsilon_m^{ln(c)})^2 = [ln(1 + f^o + \frac{a^o}{c_m^o})]^2 + [ln(1 + f^h + \frac{a^h}{c_m^h})]^2$. $f^o = 20\%$, $a^o =20\ pg/m^3$.

| Emission ($kg/hr$) | $a^h$ =10 $pg/m^3$ | $a^h$ =20 $pg/m^3$ | $a^h$ =50 $pg/m^3$ |
|---|---|---|---|
| $f^h = 0$ | 69.3 | 64.0 | 62.1 |
| $f^h = 10\%$ | 67.3 | 63.4 | 60.9 |
| $f^h = 20\%$ | 65.3 | 61.5 | 59.1 |
| $f^h = 50\%$ | 61.5 | 58.0 | 55.1 |

(Pergaud et al., 2009, QNSE), MYNN 2.5 level TKE (Nakanishi and Niino, 2006, MYNN), ACM2 (Pleim, 2007, ACM2), Bougeault and Lacarrere (Bougeault and Lacarrère, 1989, BouLac), University of Washington (Bretherton and Park, 2009, UW), Total energy mass flux (Angevine et al., 2010, TEMF), and Grenier Bretherton MaCaa (Grenier and Bretherton, 2001, GBM) schemes. Nine simulations were conducted with the PBL schemes and their associated surface layer schemes, except

for the YSU, BouLac, UW, and GBM cases in which the MM5 Monin-Obukhov surface scheme was applied. The land-surface model was Noah land-surface model (Chen and Dudhia, 2001), except ACM2 case in which Pleim-Xiu land-surface model was used.

An individual TCM is generated using each of the nine simulations. The nine TCMs can be used to estimate the emission strengths independently following the same procedure as described previously. Tables 8 and 9 show the 3rd (25th percentile),

5th (median), and 7th (75th percentile) emission strength of the nine estimates that minimize the normalized $\mathcal{F}$ defined in Equation 5 with different $f^h$ and $a^h$, while keeping $f^o = 20\%$, $a^o =20\ pg/m^3$, using concentration and logarithm concentration as the metric variables, respectively. The 25th percentile and 75th percentile values are mostly within $5\%$ of the median estimates. While the median estimates show the same trends with $f^h$ and $a^h$ as the results in Tables 6 and 7, they are significantly larger due to the meteorological model differences. Apparently the differences among the simulations with different

PBL schemes are smaller than the differences between the ensemble simulations here and the simulation used in the earlier sections. This suggests that uncertainties of the emission strength are probably larger than the ranges indicated by the 25th and 75th percentile values. The results using logarithm concentration as the metric variable are quite robust with the listed model uncertainty parameters. However, the estimates using concentration as the metric variable are very sensitive to $f^h$ and $a^h$. This is consistent with results shown in Sections 3.2 and 3.3.

Instead of using each individual TCM generated from nine simulations independently, the nine TCMs can be combined into one matrix by taking the median or average values. The combined TCM can then be used to estimate the source terms. The results for concentration and logarithm concentration metric variables are listed in Tables 10 and 11, respectively. They show that the emission estimate using the median transfer coefficients of the nine TCMs are very close to the median of the nine estimates using the nine simulations individually. For the cases with logarithm concentration as the metric variable, the

emission estimates using the median value of the nine TCMs are all within $3.1\%$ of the median values of the nine estimates obtained with each individual TCM. For the cases with concentration as the metric variable, the average relative differences




are $6.4\%$, with the maximum relative difference being $10.8\%$ when $f^h = 10\%$ and $a^h$=50 $pg/m^3$. Combining the TCMs by taking the median value generates slightly better results than combining the TCMs by taking the average value does.

Similar to what was found in earlier sections and also in Chai et al. (2015), the cases having logarithm concentration as the metric variable generally yield better results than those having concentration as the metric variable. It is probably due to

5  the large range of the concentrations. When having concentration as the metric variable, certain model uncertainty parameters yield good source terms, but the estimates are quite sensitive to the choices of the model uncertainty parameters. However, it is not easy to find such model uncertainty parameters that would yield satisfactory results for applications when the actual releases are indeed unknown. The results here and in the previous sections show that the estimates when having logarithm concentration as the metric variable are quite robust for a reasonable range of model uncertainty parameters. For these reasons,

10  logarithm concentration is chosen as the metric variable for the later tests.

**Table 8.** The 3rd (25th percentile), 5th (median), and 7th (75th percentile) emission strength of nine simulations of release 2 that minimizes the normalized $\mathcal{F}$ defined in Equation 5 for different $f^h$ and $a^h$. Concentration is taken as the metric variable. $\epsilon^2 = (f^o \times c^o + a^o)^2 + (f^h \times c^h + a^h)^2$. $f^o = 20\%$, $a^o$ =20 $pg/m^3$.

| Emission $(kg/hr)$ | $a^h$ =10 $pg/m^3$ | $a^h$ =20 $pg/m^3$ | $a^h$ =50 $pg/m^3$ |
|---|---|---|---|
| $f^h = 0$ | 6.0, 7.0, 7.2 | 7.4, 8.8, 8.8 | 13.4, 15.1, 15.3 |
| $f^h = 10\%$ | 20.0, 21.0, 21.9 | 23.9, 26.1, 27.2 | 33.2, 35.2, 37.4 |
| $f^h = 20\%$ | 48.5, 49.9, 59.1 | 53.0, 54.6, 62.8 | 58.5, 62.8, 68.6 |
| $f^h = 50\%$ | 191, 205, 274 | 186, 197, 258 | 158, 168, 207 |

**Table 9.** The 3rd (25th percentile), 5th (median), and 7th (75th percentile) emission strength of nine simulations of release 2 that minimizes normalized $\mathcal{F}$ defined in Equation 5 for different $f^h$ and $a^h$. Logarithm concentration is taken as the metric variable. $(\epsilon_m^{ln(c)})^2 = [ln(1 + f^o + \frac{a^o}{c_m^o})]^2 + [ln(1 + f^h + \frac{a^h}{c_m^h})]^2$. $f^o = 20\%$, $a^o$ =20 $pg/m^3$.

| Emission $(kg/hr)$ | $a^h$ =10 $pg/m^3$ | $a^h$ =20 $pg/m^3$ | $a^h$ =50 $pg/m^3$ |
|---|---|---|---|
| $f^h = 0$ | 102, 106, 113 | 93.4, 100, 105 | 83.8, 88.9, 97.2 |
| $f^h = 10\%$ | 97.2, 102, 108 | 88.9, 96.3, 101 | 80.5, 85.4, 94.4 |
| $f^h = 20\%$ | 93.4, 98.2, 105 | 86.3, 92.5, 98.2 | 78.1, 82.9, 91.6 |
| $f^h = 50\%$ | 88.9, 93.4, 101 | 82.9, 88.0, 94.4 | 75.8, 81.3, 87.2 |

## 3.5 Source location and other releases

In addition to the source strength, the source location and its temporal variation can be retrieved with adequate accuracy using the HYSPLIT inverse system described here if there are sufficient measurements available. For instance, Chai et al. (2015) estimated 99 6-hr emission rates of the radionuclide Cesium-137 from the Fukushima nuclear accident using 1296 daily





**Table 10.** Emission strength estimates by using average and median value of nine simulations for release 2. The cost function is normalized $\mathcal{F}$ as in Equation 5. Concentration is taken as the metric variable. $\epsilon^2 = (f^o \times c^o + a^o)^2 + (f^h \times c^h + a^h)^2$. $f^o = 20\%$, $a^o =20\,pg/m^3$.

| Emission $(kg/hr)$ | $a^h =10\,pg/m^3$ | $a^h =20\,pg/m^3$ | $a^h =50\,pg/m^3$ |
|---|---|---|---|
| $f^h = 0$ | 7.2, 7.5 | 8.9, 9.1 | 15.6, 15.9 |
| $f^h = 10\%$ | 22.3, 23.4 | 22.2, 28.0 | 37.0, 37.0 |
| $f^h = 20\%$ | 55.1, 53.0 | 59.7, 58.0 | 66.6, 64.7 |
| $f^h = 50\%$ | 213, 227 | 205, 213 | 178, 177 |

**Table 11.** Emission strength estimates by using average and median value of nine simulations for release 2. The cost function is normalized $\mathcal{F}$ as in Equation 5. Logarithm concentration is taken as the metric variable. $(\epsilon_m^{ln(c)})^2 = [ln(1+f^o+\frac{a^o}{c_m^o})]^2 + [ln(1+f^h+\frac{a^h}{c_m^h})]^2$. $f^o = 20\%$, $a^o =20\,pg/m^3$.

| Emission $(kg/hr)$ | $a^h =10\,pg/m^3$ | $a^h =20\,pg/m^3$ | $a^h =50\,pg/m^3$ |
|---|---|---|---|
| $f^h = 0$ | 115, 108 | 105, 100 | 95.3, 90.7 |
| $f^h = 10\%$ | 110, 103 | 100, 95.3 | 91.6, 87.2 |
| $f^h = 20\%$ | 105, 100 | 97.2, 92.5 | 88.9, 85.4 |
| $f^h = 50\%$ | 100, 96.3 | 93.4, 88.9 | 86.3, 82.1 |

average air concentration measurements at 115 stations around the globe. Here the system's capability to locate a single source location will be tested using a straightforward approach. In these tests, the release time is assumed known, but its location and strength are left to be determined. A region of suspect is first gridded at certain spatial resolution to form a limited number of candidate source locations. An optimal strength is then found at each candidate source location following the method described earlier. The location that results in the best match between the predicted and the observed concentrations is considered as the likely source location.

In the following tests, a $11\times11$ grid with $0.2°$ resolution in both longitude and latitude directions is used to generate 121 candidate source locations. They are centered at (40.0°N, 84.5°W) for releases 1–4, and centered at (46.6°N, 80.8°W) for releases 5 and 7. Using the normalized $\mathcal{F}$ defined in Equation 5 and assuming $f^o = 20\%$, $a^o =20\,pg/m^3$, $f^h = 20\%$, and $a^h$ $=20\,pg/m^3$, a minimal cost function associated with an optimal release strength can be found at each location. Figure 4 shows the 121 candidate locations and their respective minimal cost function values for release 2. No candidate locations are chosen to collocate with the actual source location which will be unknown for the future applications that need to locate the sources. A global minimal point is found at (39.8°N, 84.5°), with $\mathcal{F}_{min} = 3.14$ achieved when q=48.5 kg/hr. This grid point is taken as the estimated source location and it is 26.4 km away from the actual release site (39.90°N, 84.22°W). The neighboring location (39.8°N, 84.3°W) which is the closest to the actual release site yields a slightly larger $\mathcal{F} = 3.17$ with an optimal release rate of 60.9 kg/hr. If the exact source location is known as in the tests presented earlier, the cost function $\mathcal{F}$ reaches 1.59 at its minimal point when $q = 61.5$ kg/hr. Apparently, compared with those cases when the release strength is the only unknown, finding both



the source location and its strength with the same amount of observations is expected to be more difficult. Note that the smaller normalized $\mathcal{F}$ values in Figure 3 are for a case with different observation and model uncertainty parameters, where $f^o = 10\%$, $a^o = 20\ pg/m^3$, $f^h = 0\%$, and $a^h = 50\ pg/m^3$.

Table 12 lists the source location and strength estimations for the six releases following the same procedure as described
here, where the uncertainty parameters are $f^o = 20\%$, $a^o = 20\ pg/m^3$, $f^h = 20\%$, and $a^h = 20\ pg/m^3$. Releases 1 and 4 have the minimal cost function $\mathcal{F}_{min}$ occur at the north boundary and the west boundary, respectively. In such scenarios, it might be necessary to expand the suspected source region for the future applications to find the source locations. However, if source locations are known to reside in the suspected region, the sources can definitely be near the boundaries. In such cases, the point with $\mathcal{F}_{min}$ should be considered as the estimated source location. Releases 3, 5, and 7 have its $\mathcal{F}_{min}$ occurred at inner grid
points, similar to release 2 shown in Figure 4. None of the closest candidate source locations yield the best match between model simulation and observations quantified by the cost function $\mathcal{F}$. Among the six releases, the estimated source location for release 2 is the closest to its actual release site, with a distance of 26.4 km.

The release rates obtained along with the likely source locations are underestimated by a factor of 3 for release 1, and overestimated by a factor of 3 for releases 4 and 7, while the estimates for releases 2, 3, and 5 are much better, with relative
errors as $-27.6\%$, $-5.4\%$, and $21.5\%$, respectively. Table 12 also lists the release rates estimated with the exact source location assumed known. These estimates for all releases are within a factor of two compared with the actual release rates and the largest relative error is $53.3\%$ for release 1. Either with the source location known or unknown, release 2 has one of the best emission estimates among the six releases, probably because the HYSPLIT forward model has the best performance for the same release (Hegarty et al., 2013). The significant model errors when simulating the transport and dispersion even with the exact source
terms are mostly caused by the meteorological uncertainties while the HYSPLIT physical schemes and parameters, as well as the numerical discretization also contribute.

The meteorological field and the observations are the two major inputs to the current inverse modeling. As discussed above, better model performance of release 2 helps to lead to better inverse results than the other releases. However, it is impossible to eliminate the model uncertainties. In practice, ensemble runs can be used to quantify the uncertainties and reduce the model
errors by taking the average or median values of the ensemble runs. On the other hand, increasing the number of observations is effective to improve the inverse modeling results and reduce the result uncertainty. In principle, when the release strength is the only value to be determined, each measurement within the predicted plume can provide an independent estimate. However, relying on a single observation to estimate the strength is problematic since a particular model output can be very different from the observation and thus leading to an erroneous estimation of the source strength when used in isolation. For instance,
although the HYSPLIT predictions of release 2 with exact source terms are very good, compared with individual measurements, it has severe underestimation, $0.77\ pg/m^3$ predicted versus $686\ pg/m^3$ measured, as well as significant overestimation, $2033\ pg/m^3$ predicted versus $31.2\ pg/m^3$ measured. Therefore, similar to a regression technique, increasing the sampling number can improve the final results, as exemplified by the very good source term estimation for release 2 when using all the available measurements. Also note that the samples outside predicted plumes do not contribute to the inverse modeling. Table 1 lists
the total measurement counts for each release, but the number of measurements actually contributing to the inverse modeling



are those inside the HYSPLIT plumes, including those with zero or background concentrations. The number of such effective measurements inside the plumes generated by HYSPLIT from the exact source location and time period are reduced to 148, 237, 211, 68, 46, and 53, for releases 1–5, and 7, respectively. The largest number of effective measurements, 237, of release 2, also indicates the best performance of the HYSPLIT simulation among those of the six releases. The effectiveness of 5 the measurements will change when source location or release time is changed. The measurements that are not active in determining the source strength with a known source location and release time may be effective to locate the source locations.

**Table 12.** The source location (latitude, longitude) and release rate $q_{min}$ identified by the minimal normalized cost function $\mathcal{F}_{min}$ for each CAPTEX release. A total of 121 candidate locations are prescribed with $0.2°$ resolution in both longitude and latitude directions, centered at ($40.0°$N, $84.5°$W) for releases 1-4, and at ($46.6°$N, $80.8°$W) for releases 5 and 7. $\Delta$ is the distance between the point with $\mathcal{F}_{min}$ and the actual release site. $q'$ is the estimated release rate by assuming that the actual release location is known. For all the cases, $f^o = 20\%$, $a^o = 20$ $pg/m^3$, $f^h = 20\%$, and $a^h = 20$ $pg/m^3$.

| | Source location (latitude, longitude) | | $\Delta$(km) | Release rate (kg/hr) | | |
|---|---|---|---|---|---|---|
| # | Actual | Estimated | | Actual | $q_{min}$ | $q'$ |
| 1 | $39.80°$, $-84.05°$ | $41.0°$,$-83.9°$ | 134.2 | 69.3 | 23.9 | 106.3 |
| 2 | $39.90°$, $-84.22°$ | $39.8°$,$-84.5°$ | 26.4 | 67.0 | 48.5 | 61.5 |
| 3 | $39.90°$, $-84.22°$ | $40.8°$,$-85.3°$ | 135.8 | 67.0 | 63.4 | 41.7 |
| 4 | $39.90°$, $-84.22°$ | $40.2°$,$-85.5°$ | 114.1 | 66.3 | 185.7 | 75.1 |
| 5 | $46.62°$, $-80.78°$ | $46.2°$,$-81.0°$ | 49.7 | 60.0 | 72.9 | 42.6 |
| 7 | $46.62°$, $-80.78°$ | $47.4°$,$-81.2°$ | 92.5 | 61.0 | 201.0 | 66.0 |

## 4 Summary

A HYSPLIT inverse system developed to estimate the source term parameters has been evaluated using the CAPTEX data collected from six controlled releases. In the HYSPLIT inverse system, a cost function is used to measure the differences between model predictions and observations weighted by the observational uncertainties. Inverse modeling tests with various observational uncertainties show that calculating concentration differences results in severe underestimation while calculating logarithm concentrations differences results in overestimation. Introducing model uncertainty terms improves inverse results for both choices of the metric variables in the cost function. It is also found that cost function normalization can avoid spurious minimal source terms when using logarithm concentration as the metric variable. The inverse tests show that having logarithm 15 concentration as the metric variable generally yields better results than having concentration as the metric variable. The estimates having logarithm concentration as the metric variable are robust for a reasonable range of model uncertainty parameters. Such conclusions are further confirmed with nine ensemble runs where meteorological fields were generated using a different version of WRF meteorological model with varying PBL schemes.





**Figure 4.** Distribution of 121 candidate source locations for release 2. The minimal cost function at each location associated with an optimal release strength is indicated by color. The cost function defined in Equation 5 is calculated with $f^o = 20\%$, $a^o = 20\ pg/m^3$, $f^h = 20\%$, and $a^h = 20\ pg/m^3$. The actual source location , Dayton, Ohio, U.S., is shown as a red diamond.





With a fixed set of observational and model uncertainty parameters, the inverse method with logarithm concentration as the metric variable is then applied to all the six releases. The emission rates are well recovered with the largest relative error as 53.3% for release 1. The system is later tested for its capability to locate a single source location as well as its source strength. The location and strength that result in the best match between the predicted and the observed concentrations are considered as

the inverse results. The estimated location is close to the actual release site for release 2 of which the forward HYSPLIT model has the best performance. The strength estimates are all within a factor of 3 for the six releases.

*Code and data availability.* The HYSPLIT model is publicly available at https://www.ready.noaa.gov/HYSPLIT.php. The CAPTEX data can be downloaded from https://www.arl.noaa.gov/wp_arl/wp-content/uploads/documents/datem/exp_data/captex/.

*Competing interests.* No competing interests are present.

*Acknowledgements.* This study was supported by NOAA grant NA09NES4400006 (Cooperative Institute for Climate and Satellites-CICS) at the NOAA Air Resources Laboratory in collaboration with the University of Maryland.



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
