# Peer review of "Weak-constraint inverse modeling using HYSPLIT Lagrangian dispersion model and Cross Appalachian Tracer Experiment (CAPTEX) observations – Effect of including model uncertainties on source term estimation"

_Geoscientific Model Development, 2018_

## Short Comment (SC1) · 19 Jul 2018

Dear authors, in my role as Executive editor of GMD, I would like to bring to your attention our Editorial version 1.1: http://www.geosci-model-dev.net/8/3487/2015/gmd-8-3487-2015.html This highlights some requirements of papers published in GMD, which is also available on the GMD website in the 'Manuscript Types' section: http://www.geoscientific-model-development.net/submission/manuscript_types.html

[Figure]

In particular, please note that for your paper, the following requirement has not been met in the Discussions paper:

- "The main paper must give the model name and version number (or other unique identifier) in the title."

Please provide the version number of HYSPLIT in the title of your revised manuscript.

Additionally, GMD is encouraging authors to upload the program code of models (including relevant data sets) as supplement or make the code and data of the **exact model version described in the paper** accessible through a DOI (digital object identifier). In case your institution does not provide the possibility to make electronic data accessible through a DOI you may consider other providers (eg. zenodo.org of CERN) to create a DOI. Please note that in the code availability section you can still point the reader to how to obtain the newest version.

Yours, Astrid Kerkweg
* * *

---

## Referee Comment (RC1) · Anonymous Referee #1 · 29 Aug 2018

The manuscript addresses an inverse modelling study using CAPTEX data in which the effect of including model uncertainties is analyzed in the frame of source term estimation. The manuscript is well written, however, the potential contribution of the study is not clear. The study is focused on highlighting two major points: (i) advantage in using differences of logarithm of concentrations in source estimation and (ii) improvement in source estimation using a hypothetical form of observation and model uncertainty. This is not new and already established in the literature of parametric estimation problems and in the solution of inverse problems. Based on result and discussion, the study seems another application of source term estimation with sensitivity to their hypothetical model uncertainty formulation but there is no development in view of model, methodology or estimation. Following are my comments:

Major comments:

1. A major question is regarding the hypothetical form of the error formulation?. It is not clarified why this particular form is chosen?. Also, what is the evidence or guarantee that the same formulation with observed coefficients would work in other or similar source term estimation problems?. 2. The Authors did not explain well if their inverse problem is over-determined or under-determined. By noting their discretized grid and number of measurements, it seems an over-determined problem, If so, why do you need a smoothing constraint? 3. Authors did not explain why the using concentration difference and logarithmic concentration difference results so differently for the estimation of release rate. How could be the difference is so drastic between table 2 and table 3. 4. It is not clear how cost function normalization can avoid spurious solutions in logarithm concentration difference?. Does this spuriousness appear while using only concentration differences? 5. What is the utility of introducing the third part of the equation 1 if coefficient $c\_sm$ is always put to zero?. 6. The coefficients a0, f0 or ah, fh are chosen arbitrary, there is no justification why a particular set has been chosen ? 7. In Figure 2, the scale of obervations and model concentration is not clear ? Is it correct? What are the range of observed and modelled concentrations?. 8. Adding the model uncertainties to epsilon_m are simply an increase of magnitude in the previously chosen quadratic function based on observed concentration. Did you try by increasing only the values of f0 or a0 to analyze the same kind of effect?. 9. Page 9, line 30, Another aspect …...as the metric variable. What does it mean that range of release estimates are not as large as those using concentration variable ? 10. What about uncertainties in the source parameters due to varying nature of model or observation uncertainties. Is it possible to compute it with given procedure?.

---

## Referee Comment (RC2) · Anonymous Referee #2 · 4 Sep 2018

This study investigates the performance of a source term estimation method using data from the CAPTEX controlled release experiment. The interest in this experiment is that the source strength is known, as in OSSE's. However, unlike theoretical OSSE experiments real data are used, which allows assessing the role of transport model uncertainties and how to account for them. In principle this is all very interesting, however, the outcome remains on a very technical level. It is not clear what we learn here that was not known already. There is little justification of the error assumptions that

are used. In OSSE's this is fine as long as the world is self-consistent (or deliberately not), however, the use of real data calls for a justification of what is assumed. Almost no attempt is made to test whether the statistics are self-consistent (e.g. chi-squares, biased residuals, etc.). Hardly any effort is made to interpret the results: how to explain them, and to what extent are they within expectation. Furthermore, no attempt is made to relate the outcome to what was done before. These aspects will need further effort to make this manuscript suitable for publication.

Abstract, line 12, 13: To me it seems that if the problem is linear, averaging outcomes of inversions using different models should lead to the same result as using the average model for in a single inversion. Differences are then due to non-linearity (e.g. using a logarithmic cost function)

Page 5, equation 1: The smoothing part of the cost function is included but not used. In that case just leave it out.

Page 10, section 3: The explanation of how you normalize the cost function comes only at the end. To follow the discussion preceding that point it would be clearer to move it to the beginning of the section.

Table 10, 11, 12: What is missing here is an estimate of the posterior uncertainty. Otherwise there is no references to compare the actual performance to the expected performance. Without this information it is difficult to judge how important model uncertainties are. Of course, the outcome will depend on the assumed flux and observational uncertainties. However, some discussion of the validity of the assumptions regarding those is needed anyway.

Page 17, line 14-15: How significant is the finding of logarithmic inversions giving better results? Looking at your results it seems to me that they may largely be explained by a few high measurements that the model cannot really resolve at the resolution that is used. The logarithmic cost function may allow more flexibility to cope with a few "outliers". This could also explain the dependence of your results on relative observational error. Would this conclusion be different if you filter for data that the inversion has difficulty reproducing?

---

## Author Comment (AC1) · 4 Oct 2018

The HYSPLIT version number is added to the title. In addition, the version number has been explicitly given in abstract section and the main text.

The url location (https://www.ready.noaa.gov/HYSPLIT.php ) to obtain the newest version of the model, has been provided in Code and data availability part.

---

## Author Comment (AC2) · 4 Oct 2018

**General comments:**
*The manuscript addresses an inverse modelling study using CAPTEX data in which the effect of including model uncertainties is analyzed in the frame of source term estimation. The manuscript is well written, however, the potential contribution of the study is not clear. The study is focused on highlighting two major points: (i) advantage in us-*

*ing differences of logarithm of concentrations in source estimation and (ii) improvement in source estimation using a hypothetical form of observation and model uncertainty. This is not new and already established in the literature of parametric estimation problems and in the solution of inverse problems. Based on result and discussion, the study seems another application of source term estimation with sensitivity to their hypothetical model uncertainty formulation but there is no development in view of model, methodology or estimation.*

We thank the referee for thoroughly reading the manuscript and providing valuable comments. For the two major points pointed out by the referee, we try to emphasize the second point, "the effect of including model uncertainties on source term estimation", as explicitly stated in the subtitle. Although it is not new to consider model uncertainty in the inverse problems, the model uncertainties in the literature of parametric estimation problems are mostly given as static terms and they will not vary with model source terms. n source estimates This has been clarified in abstract.

In abstract, "Before introducing model uncertainty terms" has been changed to "Before introducing model uncertainty terms that depend on source estimates".

Another uniqueness of this source term estimation experiment is that the exact source terms in CAPTEX field experiment are known. So, the source term estimation methodology can be thoroughly evaluated, including "the effect of including model uncertainties on source term estimation" emphasized in this study.

**Major comments:**

1. *A major question is regarding the hypothetical form of the error formulation?. It is not clarified why this particular form is chosen? Also, what is the evidence or guarantee that the same formulation with observed coefficients would work in*

*other or similar source term estimation problems?*

The hypothetical error formulation is used mainly for its simplicity. It is clarified when the formulation is first introduced (see below). The same formulation may or may not work in other or similar source term estimation problems, but we believe that does not affect the demonstration of "the effect of including model uncertainties on source term estimation".

The following text,

*Firstly, the observational uncertainties are formulated to include a fractional component $f^o \times c^o$ and an additive part $a^o$. No model uncertainties are considered to contribute to $\epsilon$.*

has been replaced with,

*Firstly, no model uncertainties are considered to contribute to $\epsilon$. The observational uncertainties are formulated to include a fractional component $f^o \times c^o$ and an additive part $a^o$. Note that this general formulation is chosen for its simplicity. It should be replaced when more uncertainty information is available.*

2. *The Authors did not explain well if their inverse problem is over-determined or under-determined. By noting their discretized grid and number of measurements, it seems an over-determined problem, If so, why do you need a smoothing constraint?*

The smoothing constraint was not needed for the current over-determined problem. It was included to make the formulation more general. In the revised version, the smoothing term was removed from both Equations (1) and (5).

The following statement has been added at the end of Section 2.3 to clarify that all problems are over-determined.

*"Note that the cases presented in this study are all formulated as over-determined problems".*

3. *Authors did not explain why the using concentration difference and logarithmic concentration difference results so differently for the estimation of release rate. How could be the difference is so drastic between table 2 and table 3.*

The drastic differences between Table 2 and Table 3 are caused by the distinct bias directions when using concentration and logarithmic concentration in comparing model predictions and observations before introducing model uncertainty terms.

Text in the third paragraph of Section 3.1 shown below explains the cause of the significant underestimation when using concentration differences.

*The significant underestimation of the release strength is caused by the implicit assumption of a perfect model when $\epsilon$ does not include the model uncertainties. Figure 2 shows the comparison between the predicted and measured concentrations when the actual release rate of 67 $kg/hr$ is applied. Large discrepancies still exist even when the exact release is known and used in the simulation. For the measured zero concentrations, most of the predicted values are non-zero and can be above 1000 $pg/m^3$. As $\epsilon_m = a^o$ for these zero concentrations, $\frac{(c_m^h - c_m^o)^2}{\epsilon_m^2}$ will dominate the cost function when $a^o$ is not large enough. This explains that the underestimation is not as severe for $a^o$ =50 $pg/m^3$ as that for $a^o$ =10 $pg/m^3$. While $\epsilon$ do not change with $f^o$ for the zero concentrations, smaller $f^o$ values help increase the weighting of the terms $\frac{(c_m^h - c_m^o)^2}{\epsilon_m^2}$ associated with large measured concentrations. So, the estimated emission strength when $f^o = 10\%$ is better than when $f^o = 50\%$.*

The text in the same section after Equation 2 shown below explained the overestimation when using logarithmic concentration difference. It is probably not very clear and it has been revised.

*"The $\frac{a^o}{c_m^o}$ term in Equation 2 makes $\epsilon_m^{ln(c)}$ larger for measured low concentrations than those measured high concentrations, thus makes the measured zero con-*

*centrations have little effect in the final emission strength estimates".*

In the current version, it is replaced with the following.

*"The $\frac{a^o}{c^o_m}$ term in Equation 2 makes $\epsilon^{ln(c)}_m$ larger for measured low concentrations than those measured high concentrations. It causes more weighting towards measured high concentrations and results in overestimation shown in Table 3. The measured zero concentrations have little effect in the final emission strength estimates".*

4. *It is not clear how cost function normalization can avoid spurious solutions in logarithm concentration difference? Does this spuriousness appear while using only concentration differences?*

   We only see the spuriousness problems when using logarithmic concentration differences. How cost function normalization can avoid spurious solutions is illustrated in Figure 3. As explained in text, the "smaller cost function when release strength approaches zero is due to the increasing $(\epsilon^{ln(c)}_m)^2$ in Equation 4 as $c^h_m$ gets smaller". The normalization in Equation 5 makes the sum of the mismatch weighting terms constant. To make it clear, we moved Equation 5 and the rewritten preceding text shown below to the beginning of the section before introducing Figure 3.

   *To avoid having zero source as a global minimizer in such situations, the total weighted mismatch between model simulation and observations is normalized by the total weights when $q_{ij} = q^b_{ij}$, as shown in Equation 5.*

   This has been replaced with the following.

   *To avoid having zero source as a global minimizer in such situations, the sum of the weights of the mismatch between model simulation and observations is kept unchanged for varying $q_{ij}$ by normalizing it with the weight sum when $q_{ij} = q^b_{ij}$, as shown in Equation 5.*

5. *What is the utility of introducing the third part of the equation 1 if coefficient c_sm is always put to zero?.*

   This third part has been removed from both Equations 1 and 5.

6. *The coefficients a0, f0 or ah, fh are chosen arbitrary, there is no justification why a particular set has been chosen ?*

   The observation uncertainty parameters $f^o = 20\%$, $a^o$ =20 $pg/m^3$ are chosen after introducing the model uncertainty. Other $f^o$ and $a^o$ values have been tested and they show quite similar results.

   This has been clarified in Section 3.2 by adding the following in the third paragraph.

   *Additional tests with other chosen $f^o$ and $a^o$ values show similar but slightly different results. For brevity, they are not presented here.*

   A specific combination of $f^h$ and $a^h$ are not chosen until Table 12, which chooses $f^h = 20\%$, $a^h$ =20 $pg/m^3$. Tables 7, 9, and 11 have shown the results are not sensitive to the $f^h$ and $a^h$ choices. To clarify this, the following sentence has been added to the second paragraph of Section 3.5.

   *When logrithm concentration is taken as the metric variable, the emission estimates are not sensitive to $f^h$ and $a^h$ choices, as indicated by the results in Tables 7, 9, and 11,*

7. *In Figure 2, the scale of obervations and model concentration is not clear? Is it correct? What are the range of observed and modelled concentrations?*

   The largest observed and modelled concentrations are 24600 pg/m$^3$ and 25661 pg/m$^3$, respectively. For release 2, there are four observed 3-hr average concentrations above 10000 pg/m$^3$. To allow logarithm calculation, a constant 1 pg/m$^3$ is added to both observed and modelled concentrations. So zero observed and modelled concentrations appear as 1 pg/m$^3$ in the figure.

8. *Adding the model uncertainties to epsilon_m are simply an increase of magnitude in the previously chosen quadratic function based on observed concentration. Did you try by increasing only the values of f0 or a0 to analyze the same kind of effect?*

The constant part of the model uncertainties, $a^h$, has the same kind of effect as its counterpart of the observational uncertainties, $a^o$. This has been pointed out in section 3.2 when presenting results in Table 4, as shown below.

*When $f^h = 0\%$, $a^h = 10, 20$, and $50$ $pg/m^3$ while $a^o$=20 $pg/m^3$, the $q$ estimates, 7.7, 9.1, and 13.6 $kg/hr$, are inline with the results shown in Table 2, where $q = 7.1$ $kg/hr$ for $a^o$=20 $pg/m^3$ and $q = 12.6$ $kg/hr$ for $a^o$=50 $pg/m^3$.*

However, $f^h$ is applied to the predicted concentrations which vary with different source term estimation. Such "dynamic" effect of the "model uncertainty terms that will depend on source estimates" cannot be replicated with $f^o$ which applies to the "static" measurements. In fact, this is the new aspect we want to emphasize in this paper. As stated earlier, the abstract has been changed to make it clear.

9. *Page 9, line 30, Another aspect ... as the metric variable. What does it mean that range of release estimates are not as large as those using concentration variable ?*

It refers to the results in Tables 4 and 5. This has been clarified with the rewritten sentence as below.

*Another aspect of using logarithm concentration as the metric variable is that the range of the release estimates listed in Table 5 are not as large as those in Table 4 resulted from using concentration as the metric variable for the same 12 combinations of $a^h$ and $f^h$.*

10. *What about uncertainties in the source parameters due to varying nature of model or observation uncertainties. Is it possible to compute it with given procedure?*

[Figure]

The expected error $\epsilon_{q'}$ of the estimated release rate when assuming the actual release location is known has been calculated for each release. They are listed besides $q'$ as the last column in Table 12. The following text has been added to the fourth paragraph in Section 3.5.

*The posterior uncertainties of the release rate estimates $\epsilon_{q'}$ are also calculated and listed. They range from 1.8 kg/hr for release 2 to 6.2 kg/hr for release 1. The apparent underestimation is likely due to the model uncertainty assumption, including its simplified formulation as well as the chosen parameter values.*

---

## Author Comment (AC3) · 4 Oct 2018

*This study investigates the performance of a source term estimation method using data from the CAPTEX controlled release experiment. The interest in this experiment is that the source strength is known, as in OSSE's. However, unlike theoretical OSSE experiments real data are used, which allows assessing the role of transport model uncertainties and how to account for them. In principle this is all very interesting,*

*however, the outcome remains on a very technical level. It is not clear what we learn here that was not known already. There is little justification of the error assumptions that are used. In OSSE's this is fine as long as the world is self-consistent (or deliberately not), however, the use of real data calls for a justification of what is assumed. Almost no attempt is made to test whether the statistics are self-consistent (e.g. chi-squares, biased residuals, etc.). Hardly any effort is made to interpret the results: how to explain them, and to what extent are they within expectation. Furthermore, no attempt is made to relate the outcome to what was done before. These aspects will need further effort to make this manuscript suitable for publication.*

We thank the reviewer for reading the manuscript thoroughly and providing the insightful comments and constructive suggestions.

Using data from the CAPTEX controlled release experiment, it provides a unique opportunity to evaluate the source term estimation more realistically than OSSEs. In the literature of parametric estimation problems, the model uncertainties are given as static terms and they will not vary with model source terms. We try to emphasize this by adding "the effect of including model uncertainties on source term estimation", as a subtitle.

In abstract, "Before introducing model uncertainty terms" has been changed to "Before introducing model uncertainty terms that depend on source estimates".

For simplification, both observation and model errors are assumed to take the linear form and uncorrelated. This can certainly be improved in the future, they seem to be adequate to demonstrate the benefit of using the model uncertainty terms that depend on source estimates.

The self-consistency of the method has been checked. Probability density functions of $ln(c^h) - ln(c^o)$ for the six CAPTEX releases using the estimated source terms are added as Figure 5. The following paragraph is also added in Section 3.5 to justify the normal distribution assumption of $ln(c^h) - ln(c^o)$ and interpret the results.

*An assumption made in this inverse modeling algorithm is that the differences between model and observation have a normal distribution with a zero mean. Figure 5 shows the probability density function (pdf) of $ln(c^h) - ln(c^o)$ for the six CAPTEX releases using the estimated release rate $q'$ listed in Table 12. The pdf distribution of $ln(c^h) - ln(c^o)$ for Release 2 is consistent with the normal distribution assumption, and the pdf for Release 4 shows the largest deviation from a normal distribution, while those for the other four releases resembles normal distribution to some extent. The largest relative error for Release 1 is likely related to the negative mean of the $ln(c^h) - ln(c^o)$ distribution shown in Figure 5. The overestimated $q'$ probably results from the compensation of the model bias. Note that the better performance using $ln(c^h) - ln(c^o)$ than $c^h - c^o$ is believed to be caused by the fact that normal distribution assumption is mostly valid for the former but probably invalid for the latter.*

In addition, the expected error $\epsilon_{q'}$ of the estimated release rate when assuming the actual release location is known has been calculated for each release. They are listed as the last column in Table 12. The following text has been added to the fourth paragraph in Section 3.5.

*The posterior uncertainties of the release rate estimates $\epsilon_{q'}$ are also calculated and listed. They range from 1.8 kg/hr for release 2 to 6.2 kg/hr for release 1. The apparent underestimation is likely due to the model uncertainty assumption, including its simplified formulation as well as the chosen parameter values.*

To highlight the difference between this work and what was done before, the following sentence is added to the Summary section besides the change made in Abstract.

*Unlike other STE applications where model uncertainties are either ignored or assumed static, we introduce the model uncertainty terms that depend on the source term estimates.*

**Specific comments:**

[Figure]

- *Abstract, line 12, 13: To me it seems that if the problem is linear, averaging outcomes of inversions using different models should lead to the same result as using the average model for in a single inversion. Differences are then due to non-linearity (e.g. using a logarithmic cost function)*

We agree with the reviewer's statement on the linear systems. As the referee pointed out, logarithmic cost function will result in non-linearity. For the current inverse system that minimizes the cost function with a background term, the average of inversion results using two different models are not identical to the inversion results of using the average of the two model TCMs even without logarithmic concentration differences in the cost function.

- *Page 5, equation 1: The smoothing part of the cost function is included but not used. In that case just leave it out.*

We removed the smoothing term from both Equations 1 and 5.

- *Page 10, section 3: The explanation of how you normalize the cost function comes only at the end. To follow the discussion preceding that point it would be clearer to move it to the beginning of the section.*

Following this suggestion, we moved Equation 5 and the rewritten preceding text shown below to the beginning of the section before introducing Figure 3.

*To avoid having zero source as a global minimizer in such situations, the sum of the weights of the mismatch between model simulation and observations is kept unchanged for varying $q_{ij}$ by normalizing it with the weight sum when $q_{ij} = q_{ij}^b$, as shown in Equation 5.*

- *Table 10, 11, 12: What is missing here is an estimate of the posterior uncertainty. Otherwise there is no references to compare the actual performance to the expected performance. Without this information it is difficult to judge how important*

*model uncertainties are. Of course, the outcome will depend on the assumed flux and observational uncertainties. However, some discussion of the validity of the assumptions regarding those is needed anyway.*

The posterior uncertainty, $\epsilon_{q'}$, has been calculated for each release and they are listed as the last column in Table 12. The following discussion has been added to the fourth paragraph in Section 3.5.

*The posterior uncertainties of the release rate estimates $\epsilon_{q'}$ are also calculated and listed. They range from 1.8 kg/hr for release 2 to 6.2 kg/hr for release 1. The apparent underestimation is likely due to the model uncertainty assumption, including its simplified formulation as well as the chosen parameter values.*

- *Page 17, line 14-15: How significant is the finding of logarithmic inversions giving better results? Looking at your results it seems to me that they may largely be explained by a few high measurements that the model cannot really resolve at the resolution that is used. The logarithmic cost function may allow more flexibility to cope with a few "outlines". This could also explain the dependence of your results on relative observational error. Would this conclusion be different if you filter for data that the inversion has difficulty reproducing.*

This finding of logarithmic inversions giving better results is not new. Chai et al. (2015) has more discussion on the choice of both control variables and metric variables using "twin experiment" settings. Figure 2 shows that there are no apparent "outlines" when the exact release terms are applied in the HYSPLIT simulation. We believe that the reason the logarithmic inversion works better is due to the large range of the concentrations and the log-normal distribution of the concentration differences between model predictions and observations. The newly added Figure 5 and associated paragraph mentioned earlier have more explanation on this.

*Chai, T., Draxler, R., and Stein, A.: Source term estimation using air concen-*

*tration measurements and a Lagrangian dispersion model - Experiments with pseudo and real cesium-137 observations from the Fukushima nuclear accident, Atmos. Environ., 106, 241–251, 2015.*

---

## Author Comment (AC4) · 4 Oct 2018

The revised version is included here.

Please also note the supplement to this comment:
https://www.geosci-model-dev-discuss.net/gmd-2018-159/gmd-2018-159-AC4-supplement.pdf

[revised manuscript text omitted]